# Population Diversity and Genetic Structure Reveal Patterns of Host Association and Anthropogenic Impact for the Globally Important Fungal Tree Pathogen *Ceratocystis manginecans*

**DOI:** 10.3390/jof7090759

**Published:** 2021-09-15

**Authors:** Feifei Liu, Tuan A. Duong, Irene Barnes, Michael J. Wingfield, Shuaifei Chen

**Affiliations:** 1China Eucalypt Research Centre (CERC), Chinese Academy of Forestry (CAF), Zhanjiang 524022, China; feifei.liu@fabi.up.ac.za; 2Department of Biochemistry, Genetics and Microbiology, Forestry and Agricultural Biotechnology Institute (FABI), University of Pretoria, Pretoria 0028, South Africa; tuan.duong@fabi.up.ac.za (T.A.D.); irene.barnes@fabi.up.ac.za (I.B.); mike.wingfield@fabi.up.ac.za (M.J.W.)

**Keywords:** disease control, emerging disease, pathogen diversity, pathogen movement

## Abstract

Species in the *Ceratocystis manginecans* complex are important fungal pathogens of plantation trees globally. The most important hosts include species of *Eucalyptus*, *Acacia*, *Mangifera*, and *Punica*. Despite their relevance and widespread occurrence, little is known regarding their population genetics and how this might relate to their host associations or geographic regions in which they occur. A global collection of 491 isolates representing the *C. manginecans* complex, from four different plant hosts and nine countries, were genotyped using microsatellite markers. Population genetic analyses using numerous tools were conducted to interrogate how their genetic diversity and structure might be affected by host or areas of occurrence. Results of genetic diversity studies showed that when grouping isolates into populations based on their host associations, the population on *Eucalyptus* was most diverse, and it also has a broad global distribution. When considering countries of origin as a basis for defining populations, the gene and genotypic diversity were highest in populations from China, Indonesia, and Brazil. In contrast, populations from Oman and Pakistan collected from *Mangifera* had the lowest genetic diversity and were clonal. Molecular variance, population differentiation, and network and structure analyses showed that the genetic structure of isolates in the *C. manginecans* complex is influenced by both host association as well as geographical isolation. Furthermore, the results reflected the movement of genotypes between plant hosts and geographic regions that have implications regarding the broad global distribution of this pathogen.

## 1. Introduction

Tree diseases caused by fungal pathogens are increasing in incidence and impact globally. This is especially true in the case of trees grown for plantation forestry and fruit tree production [1,2,3,4]. As international trade and travel grows, and against a background of climate change, the emergence of biological invasions and the spread of existing fungal pathogens is inevitable [4,5,6,7,8]. Moreover, planted forests and orchards of fruit trees are typically comprised of genetically uniform monocultures, which also makes them prone to serious damage by pathogens [9,10,11]. This is illustrated by non-native tree species that have been extensively established in plantations during the course of the last two centuries, often resulting in serious damage by introduced fungal pathogens or pathogens native to the areas where they have been planted [8,12,13]. It is consequently important to understand the origins, distribution, and introduction pathways of these tree pathogens in order to manage the real or potential economic impact [3,14,15].

*Ceratocystis* spp. are amongst the most important causal agents of emerging tree diseases [16,17,18]. These pathogens cause a variety of diseases including cankers and vascular wilts [16,19]. Important examples include *C. albifundus*, the causal agent of a canker and wilt disease of *Acacia mearnsii* in Africa [20,21], *C. manginecans*, associated with wilt disease of *Mangifera indica* in Pakistan and Oman [22], *C. platani*, the causal agent of canker stain and die-back of *Platanus* spp. in Europe and the USA [23], and *C. lukuohia* and *C. huliohia*, which are associated with the rapid ‘ōhi’a death of native *Metrosideros polymorpha* trees in Hawaii [18]. Diseases such as these have led to significant economic losses to planted forests as well as the devastation of natural forest ecosystems in many countries of the world [18,24,25].

*Ceratocystis manginecans* and *C. eucalypticola* have been reported to cause serious diseases on several plant hosts, including *Acacia mangium* [25,26,27], *Eucalyptus* spp. [28,29,30], *Mangifera indica* [22,27,31,32,33,34], *Prosopis cineraria* [27,35], and *Punica granatum* [36]. *Ceratocystis manginecans* was first described from Oman and Pakistan causing a devastating wilt disease on mango trees [22], and where it was also found on the native legume trees *Prosopis cineraria* and *Dalbergia sissoo* [35]. *Ceratocystis acaciivora* was described as the causal agent of wilt and die-back disease of the legume tree *Acacia mangium* used to establish forest plantations in Indonesia [25]. This species was later reduced to synonymy with *C. manginecans* based on additional multi-gene phylogenetic analyses [37]. *Ceratocystis eucalypticola*, a species closely related to *C. manginecans,* was first described from *Eucalyptus* in South Africa [29]. In the same study, it was also shown that isolates of *Ceratocystis* from *Eucalyptus* trees in Brazil, Indonesia, Congo, Thailand, Uganda, and Uruguay were closely related to this species and most likely represent the same species [29].

*Ceratocystis manginecans* and *C. eucalypticola* are phylogenetically closely related species residing in the Latin American Clade and for which species boundaries remain poorly defined. This is largely due to a paucity of informative taxonomic markers to clearly define species in this clade and the presence of multiple ITS types in single isolates of *C. manginecans* and *C. eucalypticola* [30,32,36,37]. As a result, some authors have considered differences in the ITS region to represent different genotypes in populations and suggested that isolates of *C. manginecans* and *C. eucalypticola* should be treated as the same species [17,36,38]. In another study, Fourie et al. [37] applied phylogenetic species concept using data from multiple gene regions and showed that *C. eucalypticola* and *C. manginecans* reside in two discrete clades, justifying their recognition as distinct species. Based on the results of previous studies [17,36,37,38], and for the purpose of the present investigation, we refer to these two closely related species as members of the *C. manginecans* complex.

Several population genetic studies have been conducted on isolates of the *C. manginecans* complex [27,33,36,39]. For example, a recent study by Al Adawi et al. [33] showed that the pathogen in Oman and Pakistan was an introduced population with isolates representing a clonal linage, which is most likely from South East Asia [25] or South America [36]. The study of Fourie et al. [27] using populations of *C. manginecans* from Oman, Pakistan, Indonesia, and Vietnam suggested that *C. manginecans* could be native to South East Asia. This was based on a significantly higher genetic diversity observed amongst isolates from that area, especially in Vietnam. A very low genetic diversity was observed on Chinese populations from *Punica*, indicating that this pathogen has experienced a severe genetic bottleneck [36,39]. That study also suggested that the Chinese population on *Punica* was indirectly derived from introductions of a population on *Eucalyptus* from Brazil [36].

Although several population genetic studies have been conducted on isolates residing in the *C. manginecans* complex, these have mostly considered isolates from a limited number of plant hosts and regions. As a consequence, knowledge regarding the global population genetic diversity, genetic structure, and distribution of isolates in this complex is limited. Thus, the aims of this study were (i) to investigate the genetic diversity of the *C. manginecans* complex using a large collection of isolates from nine countries and four plant hosts using 10 polymorphic microsatellite markers, (ii) to consider whether there are patterns of population genetic structure reflecting geographic isolation, host association, or species boundaries of the isolates, and (iii) to determine possible patterns of distribution for isolates from different plant hosts or geographic regions.

## 2. Materials and Methods

### 2.1. Fungal Isolates and DNA Extractions

Isolates representing the *C. manginecans* complex (i.e., either *C. manginecans* or *C. eucalypticola*) were obtained from various countries and sources during the period 2003 to 2014 (Figure 1; Appendix A). Isolates from China were collected from the stumps of freshly harvested *Eucalyptus* trees in five provinces (FuJian, GuangDong, GuangXi, HaiNan, and YunNan provinces) and from diseased stem tissue from *Punica*
*granatum* trees in SiChuan and YunNan provinces. In YunNan province, the isolates from *Punica* and *Eucalyptus* were collected approximately 200 km from each other (Appendix A). Additional isolates were obtained from the Culture Collection (CMW) of the Forestry and Agricultural Biotechnology Institute (FABI), and these were from *Eucalyptus* spp. and *Acacia* spp. from several countries (Brazil, Congo, Indonesia, Malaysia, South Africa, and Uruguay), which were collected and used in previous studies [27,29] (Figure 1; Appendix A). In addition, 131 isolates of *C. manginecans* from *Mangifera indica* and *Acacia* spp. used in the study of Fourie et al. [27], and for which prior SSR genotyping data were already available, were also included in the analyses (Appendix A).

Single hyphal tip cultures generated for all isolates used in this study were subjected to DNA extraction and SSR genotyping. Isolates were grown on 2% Malt Extract Agar (MEA: 20 g/L malt extract from Biolab and 20 g/L agar from Difco) for 2–3 weeks at 25 °C. Mycelium was scraped from the surfaces of the agar using a sterile scalpel and transferred to 2 mL Eppendorf tubes. Genomic DNA was extracted using a modified cetyl trimethyl ammonium bromide (CTAB) protocol described by Möller et al. [40]. DNA concentrations were determined using a NanoDrop^®^ ND-1000 Spectrophotometer (Nano Drop Technologies, Wilmington, DE, USA). DNA samples were diluted in 10 mM Tris-HCl (pH 8.0) to a final concentration of approximately 50 ng/μL and stored at −20 °C until being used.

### 2.2. Confirmation of Isolate Identity

Isolates utilized in the study of Fourie et al. [27] had previously been identified as *C. manginecans* based on molecular data. The remaining isolates, newly collected for the present study, were identified based on DNA sequence analyses. These included sequences of the ITS region using primers ITS1 and ITS4 [41], the *MS204* gene region using primers MS204F.ceratoB and MS204R.ceratoB [37], and the *RPBII* gene region using primers RPB2-5Fb and RPB2-7Rb [37].

The PCR reaction mixtures and thermal cycles for all three gene regions were the same as those described by Liu et al. [42]. PCR products were cleaned with ExoSap-IT^TM^ PCR Product Clean-up Reagent (Thermo Fisher Scientific, Waltham, MA, USA) to remove unincorporated primers and dNTPs, which was followed by sequencing with the BigDye Terminator v3.1 sequencing premix kit (Applied Biosystems, Foster City, CA, USA). An ABI PRISM™ 3100 Autosequencer (Applied BioSystems, Foster City, CA, USA) was used for sequencing.

Raw sequence data were trimmed and assembled using Geneious v. 7.0; then, they were aligned using MAFFT v. 7 [43] and then manually inspected and trimmed at both terminal ends in MEGA v. 7 [44]. Maximum likelihood (ML) phylogenetic analyses were conducted for each gene dataset as well as for the combined dataset for *MS204* and *RPBII* gene regions, using the program PhyML v. 3.1 [45]. Confidence levels for the nodes were determined using 1000 bootstrap replicates. *Ceratocystis pirilliformis* (CMW 6579) was used as the out-group taxon.

### 2.3. Microsatellite Marker Genotyping and Allele Scoring

Ten microsatellite markers, specifically developed for *C. manginecans* [27], were used to genotype isolates. The PCRs were performed in five multiplex reactions as described by Fourie et al. [27], with minor adjustments to the annealing temperatures. The fragment sizes of the microsatellite amplicons were analyzed using an ABI PRISM^TM^ 3500xl Autosequencer (Thermo Fisher Scientific, Carlsbad, CA, USA). The GeneScan 500 LIZ molecular size standard (Applied Biosystems, Thermo Fisher Scientific, Carlsbad, CA, USA) was used as the internal size standard. GeneMapper^®^ v. 5 software (Applied Biosystems, Thermo Fisher Scientific, Carlsbad, CA, USA) was used for allele scoring. Different fragment sizes were considered to represent different alleles. The combination of alleles from all genotyped markers for each isolate was presented as a multi-locus genotype (MLG).

One to two representative isolates for every allele determined from fragment size analysis were selected for Sanger sequencing to confirm the allele size obtained with GeneScan. Collectively, 85 isolates were selected, and PCRs and sequencing reactions were conducted using unlabeled microsatellite primers. The sequences obtained from the forward and reverse primers were assembled in Geneious v. 7. Sequence data for each of the observed alleles from all markers were deposited in GenBank.

### 2.4. Population Genetic Diversity Analyses

To calculate the various population statistics, isolates were divided into populations based on countries of origin. Additionally, in countries (China and Indonesia) where isolates were obtained from more than one host, these were also divided into sub-populations by taking both country of origin and host plants into consideration. This made it possible to consider the relationship between genetic diversity and host associations where possible. The total number of alleles (Na), the number of effective alleles (Nef) [46], as well as the number of multilocus genotypes (MLG) were calculated using GenAIEx v. 6.5 [47]. The R package poppr [48] was used to compute the genotypic diversity, including the number of expected MLG at the smallest sample size based on rarefaction (eMLG), the Stoddart and Taylor Index of MLG diversity (G) [49], and unbiased gene diversity (Hexp) [50].

### 2.5. Population Molecular Variance, Population Differentiation, and Gene Flow

Analysis of molecular variance (AMOVA) was investigated in GENALEX v. 6.5 [47] to test the hypothesis of population differentiation among and within populations based on host and geographic origin. A dataset that considered both the countries of origin and host plants populations was prepared, and this was used to compute pairwise comparisons of population differentiation (φPT) and gene flow (Nm). This was also conducted in GENALEX v. 6.5 [47].

### 2.6. Minimum Spanning Network Analyses

To investigate possible evolutionary relationships among the observed multilocus genotypes (MLGs), minimum spanning networks (MSN) were constructed using Bruvo’s distance with the R packages poppr and ape [48,51].

### 2.7. Analyses of Population Subdivision

The model-based Bayesian clustering approach, used to study genetic differences in populations, was implemented in STRUCTURE v. 2.3.3. This detected the most likely number of sub-populations (K), based on allele frequencies per locus [52,53]. The full dataset of 491 isolates was subjected to STRUCTURE analyses with 20 independent runs with the K value ranging from 1 to 10; 1,000,000 Markov chain Monte Carlo (MCMC) iterations following a burn-in period of 250,000 iterations; and using an admixture ancestry model and an independent allele frequency model [52]. The results of the STRUCTURE analysis were submitted to STRUCTURE HARVESTER [54]. Consequently, the K value was determined based on the *△K* and the median value of lnPr (K) generated from STRUCTURE HARVESTER [54].

Since STRUCTURE tends to detect the uppermost level of genetic differentiation within a given dataset [55], subsequent STRUCTURE analyses were performed using the sub-datasets compiled from each of the clusters emerging from the initial STRUCTURE run. This made it possible to detect the presence of sub-structure in our dataset [56]. These subsequent STRUCTURE analyses were conducted using the same parameters as described for the initial STRUCURE run.

### 2.8. Mode of Reproduction

Mode of reproduction was investigated using multilocus linkage disequilibrium in the form of the Index of Association (I_A_) statistic [57]. rBarD was inferred to assess whether the populations were randomly mating (rBarD close to or equal to 0) or not (rBarD significantly greater than zero). The significance of rBarD at *p* < 0.05 was assessed by comparing the observed rBarD value with that obtained from 999 randomization samplings of the same dataset. Clone-corrected datasets were prepared and used for this calculation using the ia function in the poppr R package [48].

## 3. Results

### 3.1. Fungal Isolates and Identification

All isolates obtained from nine countries (Figure 1) used in this study had morphological characters typical of *Ceratocystis* species. DNA was successfully extracted from all isolates. PCR amplification and sequencing of the ITS, *MS204*, and *RPBII* gene regions yielded products of 610, 970, and 1200 base pairs, respectively. Some isolates had multiple ITS types [37,58], and consequently, this gene region was excluded from the combined phylogenetic analyses. As a result, the *MS204* and *RPBII* regions were used to compile the combined dataset (Figure 2). Based on the multi-gene phylogenetic analyses, 491 isolates were assigned to the *C. manginecans* complex, which included isolates that have previously been labeled as either *C. manginecans* or *C. eucalypticola*. An ML phylogenetic tree containing isolates representing all sequence haplotypes of *MS204* and *RPBII* is presented in Figure 2.

Collectively, data for a total of 491 isolates of the *C. manginecans* complex representing nine geographic regions (Brazil, China, Congo, Indonesia, Malaysia, Oman, Pakistan, South Africa, Uruguay) and occurring on four hosts (*Acacia* spp., *Eucalyptus* spp., *Mangifera indica*, and *Punica granatum*) were included in the analyses (Figure 1; Appendix A). Of these, 207 were collected from *Eucalyptus* spp. and *Punica granatum* in China in this study, and 153 isolates from *Eucalyptus* spp. and *Acacia* spp. from Brazil, Congo, Indonesia, Malaysia, South Africa and Uruguay were retrieved from the Culture Collection (CMW). In addition, data generated (using the same microsatellite markers) for 131 isolates of *C. manginecans* from *Acacia* spp. and *Mangifera indica* by Fourie et al. [27] were included in the analyses (Figure 1; Appendix A).

### 3.2. Microsatellite Amplification

All ten microsatellite markers utilized in this study were polymorphic with the number of alleles per locus ranging from two to 14. A total of 50 different alleles were generated with the ten markers. The sequencing of all alleles observed in this study from 85 representative isolates (one or two isolates per allele were selected for sequencing; Appendix A) confirmed that the variation observed between alleles was due to variation in microsatellite repeat units and not due to indels. In some cases, the sequence length differed from that obtained with fragment analyses. Consequently, all allele sizes were adjusted according to the sequence lengths obtained prior to subsequent analyses. The original and adjusted allele sizes are presented in Appendix A, respectively.

### 3.3. Population Genetic Diversity Analyses

For the geography-associated populations, gene diversity (Hexp) values obtained for nine geographic populations ranged from 0 to 0.351 (Table 1). The highest level of gene diversity (Hexp = 0.351) was observed for the population from China followed by that from Indonesia (Hexp = 0.335) and Brazil (Hexp = 0.280). The populations from Oman and Pakistan had the lowest gene diversity (Hexp = 0). Similarly, the population from China had the highest values in terms of other computed population statistics such as the number of alleles (Na) and multilocus genotypes (MLG; Table 1). The population from China also had the highest level of genotypic diversity (G = 11.528), followed by populations from Brazil (G = 8.048) and Indonesia (G = 7.399), which had similar genotypic diversity values, while populations from Oman and Pakistan were represented by only one MLG and thus had no genotypic diversity (Table 1). In addition, there were more genotypes from *Eucalyptus* (55 MLGs) than that from *Acacia* (30 MLGs), *Punica* (8 MLGs), and *Mangifera* (1 MLG).

For the sub-populations in China and Indonesia, the gene diversities (Hexp) and genotype diversity (G) of China *Eucalyptus* population were higher compare with Indonesia *Acacia*, Indonesia *Eucalyptus*, and China *Punica* populations (Table 1). For China specifically, both the gene and genotype diversity for the *Eucalyptus* population was substantially higher than for the *Punica* population (Table 1). Likewise for Indonesia, the gene diversity of the *Eucalyptus* population was higher than that from *Acacia*, while the genotype diversity of the *Eucalyptus* population was lower than that for the *Acacia* population (Table 1). In these populations, one MLG was shared between the China *Eucalyptus*, China *Punica*, and Indonesia *Eucalyptus* populations, one MLG was shared between the China *Eucalyptus* and Indonesia *Eucalyptus* populations, one MLG was shared between the Indonesia *Acacia* and Malaysia *Acacia* populations, and one MLG was shared between the China *Eucalyptus* and South Africa *Eucalyptus* populations (Figure 3).

### 3.4. Population Molecular Variance, Population Differentiation, and Gene Flow

AMOVA revealed significant genetic differentiation among populations based on the nine geographical areas and the four host plants (*p* ≤ 0.001). Based on geographic origin, a higher genetic differentiation was detected among populations (56%) and lower variation within populations (44%). In terms of host, of the total molecular variation, 51% of the genetic difference was observed among populations and 49% was within populations (Table 2).

Analysis of the population differentiation based on pairwise comparisons of population differentiation (φPT) and gene flow (Nm) (Table 3) showed that the highest levels of genetic differentiation (φPT = 0.943) were between the population from *Eucalyptus* in Uruguay and the *Mangifera* in Oman. In contrast, the lowest levels of genetic differentiation (φPT = 0.135) with highest gene flow (Nm = 3.209) were recorded between the population on *Eucalyptus* in China and the population on *Eucalyptus* in South Africa, followed by *Eucalyptus* in Uruguay (φPT = 0.178, Nm = 2.305). These results also showed that the genetic differentiation between the China *Eucalyptus* population and the *Eucalyptus* populations from other countries (φPT = 0.135 to 0.361) were all lower than that between China *Eucalyptus* population and populations from other countries that were not from *Eucalyptus* (φPT = 0.524 to 0.648) (Table 3). These results suggest that hosts play a role in the genetic differentiation of the *C. manginecans* complex.

### 3.5. Minimum Spanning Network Analyses

The minimum spanning network (MSN) analyses revealed two distinct clusters for the 491 isolates residing in the *C. manginecans* complex (Figure 3a,b). These clusters were separated by a large genetic distance and represented allelic differences at multiple microsatellite loci (Figure 3a,b). Based on the MSN, both of these genetic clusters showed that the *Mangifera* population is closely related to the *Acacia* population and the *Punica* population is closely related to *Eucalyptus* population (Figure 3a,b). In addition, the population from *Eucalyptus* in different countries, despite their being separated by large geographic distance, were genetically more closely related to one another than to populations from other hosts, with the exception of *Punica* (Figure 3b).

### 3.6. Analyses of Population Subdivision

STRUCTURE analysis with no a priori assumptions of structure for all 491 isolates suggested K = 2 as the best number of clusters for the dataset investigated (Figure 4). Similar to the distance based and MSN analyses where the isolates were also clearly distinguished by two main clusters. A total of 217 isolates were assigned to cluster 1, which included all the isolates from *Acacia* (165) and *Mangifera* (21), and 31 isolates originally collected from *Eucalyptus* in China and Indonesia (Figure 4). The remaining 274 isolates assigned to cluster 2 were all from *Eucalyptus* (194) and *Punica* (80) (Figure 4).

In the subsequent STRUCTURE analyses conducted on isolates from each of the two clusters identified in the first structure run, the optimal K values of 2 and 4 were suggested for cluster 1, while the optimal K values of 2 and 5 were suggested for cluster 2 (Figure 4). These finer-scale sub-structures were consistent with the geographic origins of isolates. Thus, the uppermost level of structure reflected host association, while the sub-structure analyses reflected the geographic origins of isolates.

### 3.7. Mode of Reproduction

Index of association values (Appendix A) were calculated from the clone-corrected datasets for each population defined by country of origin (Appendix A–f), except for the Congo, Oman, and Pakistan due to the low number of multilocus genotypes (MLGs) in these three countries (two MLGs for Congo, and one MLG for Oman and Pakistan). Populations from Brazil (rBarD: 0.062; P: 0.135; Appendix A), Malaysia (rBarD: −0.041; P: 0.939; Appendix A), South Africa (rBarD: −0.097; P: 1.000; Appendix A), and Uruguay (rBarD: 0.032; P: 0.732; Appendix A) all had observed rBarD values not significantly larger than zero, suggesting that these populations are outcrossing with respect to *Ceratocystis*. China (rBarD: 0.094; P: 0.001; Appendix A) and Indonesia (rBarD: 0.080; P: 0.001; Appendix A) populations had rBarD significantly larger than zero, suggesting that these two populations are primarily selfing or reproducing asexually.

Both the China and Indonesia populations included two sub-populations based on host. Consequently, we conducted a second analysis of the index of association based on separated host populations for these two countries (Appendix A–j). When single hosts were considered, populations from *Punica* in China (rBarD: −0.064; P: 0.845; Appendix A) and *Eucalyptus* in Indonesia (rBarD: 0.114; P: 0.01; Appendix A) showed evidence for recombination, whereas the population from *Eucalyptus* in China (rBarD: 0.110; P: 0.001; Appendix A) and *Acacia* in Indonesia (rBarD: 0.106; P: 0.002; Appendix A) was under linkage disequilibrium.

## 4. Discussion

A large collection of isolates representing the important and widely distributed *C. manginecans* complex as defined at the outset of this study from nine countries and four hosts were genotyped and analyzed using ten microsatellite markers. The results showed that the population on *Eucalyptus* was the most diverse, also having a broad global distribution. This was followed by populations from *Acacia* and *Punica*, and lastly by *Mangifera*, where populations were clonal and represented by only one MLG. Populations from China, Indonesia, and Malaysia displayed higher levels of genetic diversity than those from other countries. Broadly, the genetic structure of isolates residing in the *C. manginecans* complex was influenced by both host association and geographic isolation. Furthermore, the results reflected frequent movement of genotypes between different plant hosts and geographical regions.

Population sub-division for isolates in the *C. manginecans* complex was shown to be associated with a pattern of host association. This is evident by the greater genetic differentiation observed between two main clusters identified in both structure and MSN analyses. One of these clusters consisted of mostly isolates from *Eucalyptus* and *Punica*, whereas the other consisted of isolates from *Acacia* and *Mangifera*. The host-associated pattern was also supported by the fact that individuals assigned into the same cluster were collected from the same host tree species at different locations and separated by large geographic distances, such as in the case of *Eucalyptus* isolates from China and South Africa.

Of the four hosts considered in the current study, the population on *Eucalyptus* was most diverse. Isolates from *Eucalyptus* were also geographically broadly represented, occurring in six of the nine countries sampled. As *Eucalyptus* is not native to any of the areas considered and is a host that has been moved around globally, this could present a pathway of pathogen distribution [8,13,59]. Notably, the results also suggested the frequent and most probably recent movement of genotypes between populations on *Eucalyptus* from different countries and regions. This was supported by the sharing of MLGs between the China and South Africa populations as well as between China and Indonesia populations. In addition, there was very low population differentiation and high gene flow between the China/South Africa and China/Indonesia populations.

The results of this study indicated that the *C. manginecans* population from *Punica* was most closely related to that from *Eucalyptus*. This was also supported by the fact that there were two MLGs and 14 alleles shared between the *Eucalyptus* isolates and *Punica* isolates in the China population. All of these lines of evidence suggest that the *C. manginecans* population might have undergone recent host range expansion events.

Host range expansion events in the *C. manginecans* complex have been suggested in previous studies [27,33,36]. The single clonal genotype of *C. manginecans* that occurs on *Mangifera* in Oman and Pakistan was also found on the native legume trees, *Prosopis cineraria* (Ghaf) in Oman and *Dalbergia sissoo* (Shisham) in Pakistan [35]. These authors suggested that the fungus is an introduced pathogen in those countries and has undergone a host range expansion from *Mangifera* to the native hosts, but the source of the introduction onto *Mangifera* was unknown. The authors further suggested that this could have come from South East Asia or South America [27,33]. The later discovery of *C. manginecans* (originally as *C. acaciivora*) on *Acacia mangium*, which, similar to *Prosopis cineraria* and *Dalbergia sissoo* is a legume, suggests a host link between the diseases caused by *C. manginecans* in the Middle East and South East Asia. The results of the present study add further support for the hypothesis that *C. manginecans* on *Mangifera* is linked to the disease caused by *C. manginecans* on *Acacia* in South East Asia.

The genetic association between isolates of *C. manginecans* on *Punica* and *Eucalyptus* emerging from this study is as intriguing as that linking *C. manginecans* from the legume hosts. *Eucalyptus* residing in the Myrtaceae (Myrtales) is native to Australia and some nearby islands but has been widely planted globally for plantation development for more than 100 years [8,13,59]. There are many examples of pathogens on *Eucalyptus* having been moved globally, such as from the native area to the new environment [8,60,61]. However, there are also numerous examples of pathogens of native trees in the Myrtales having undergone host range expansion to infect *Eucalyptus* where these trees have been established as non-natives [62,63]. Notable examples of these are pathogens residing in the Cryphonectriaceae [64,65]. Thus, a host range expansion from *Eucalyptus* to *Punica* in the case of *C. manginecans* seems very probable given the fact that *Punica* is also a member of the Myrtales [66]. This might also be supported by the study of Harrington et al. [36] showing that *C. manginecans* on *Punica* could have originated from *Eucalyptus*.

The fact that some genotypes of the *C. manginecans* complex were found in isolates from more than one geographic location provides evidence that this pathogen has been accidently moved globally. Moreover, some introductions have apparently occurred recently. It is well-recognized that *Ceratocystis* species are well suited to long-distance dispersal by insects and human activities [23,67]. The introduction of these fungi into new areas can dramatically impact on local host populations [20,21,68], and this appears to have been the case for isolates in the *C. manginecans* complex. Thus, the emergence of devastating diseases of *Acacia* in Asia [25,26], *Eucalyptus* in Congo [29,69], *Punica* in China [36], and *Mangifera* in Oman and Pakistan [32,33,35] all appear to be associated with introductions of an aggressive pathogen with a strong link to a source population, apparently on *Eucalyptus*. The origin of this source population has yet to be determined. The answer to this question is complicated by the fact that there is also evidence of hybridization in *Ceratocystis* spp. (including *C. manginecans*) and possibly horizontal gene transfer [36,70,71].

Although the available cultures from different populations considered in this study varied in number, intriguing patterns of global spread emerged. There remains substantial opportunity to expand the knowledge gained in this study with larger numbers of isolates from areas or hosts that could not be sampled or where sampling was not optimal. Some genotypes occurred only on specific hosts. The results of this study suggest that host-associated genotypes should be taken into consideration in defining quarantine measures and in the assessments of invasive risks of these fungi. The two closely related species, *C. manginecans* and *C. eucalypticola*, are currently recognized based on multiple gene phylogenic analyses. Their species boundaries become unclear when a large number of isolates are included in analyses. The existence of two major *Eucalyptus*- (viz. *C. eucalypticola*) and *Acacia*- (viz. *C. manginecans*) associated clusters provides an indication of an ongoing process of speciation driven by host associations in these fungi. The results provide a foundation for future studies on genetic variation of other globally distributed populations in the *C. manginecans* complex. Collectively, these should lead to a better understanding of the genetic relationships and pathways of introduction of *Ceratocystis* spp. Furthermore, the results will contribute to managing future invasions resulting from different genotypes of the *C. manginecans* complex on different host tree species in different countries and regions.

## Figures and Tables

**Figure 1 jof-07-00759-f001:**
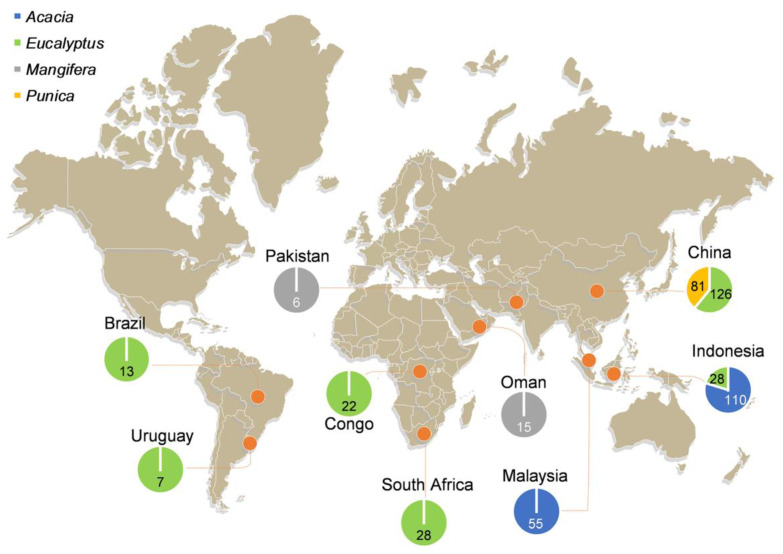
Sampling locations of *Ceratocystis* isolates used in this study. A total of 491 isolates were obtained from nine geographical locations. Pie graphs indicate the proportion of samples from each host, with the number inside the pie graphs indicating the total number of isolates from each host at that location.

**Figure 2 jof-07-00759-f002:**
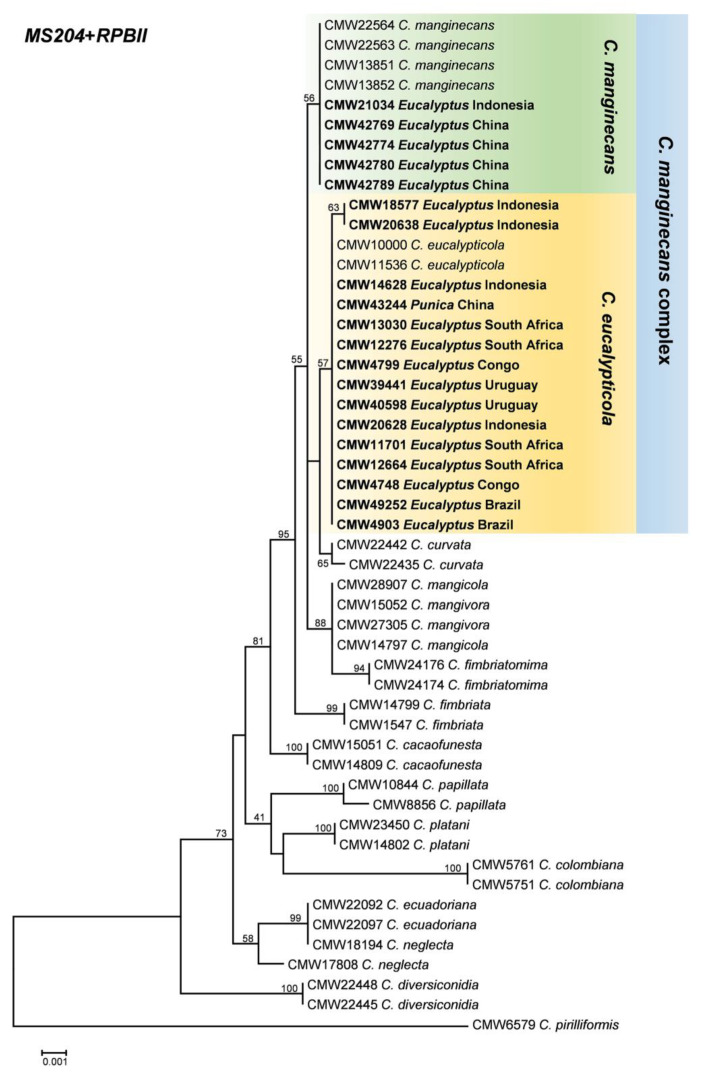
Phylogenetic tree based on maximum likelihood (ML) analysis of a combined dataset of *MS204* and *RPBII* gene sequences for *Ceratocystis* isolates used in this study (only representative haplotypes per country and host were included in the analysis). Isolates in bold and highlighted in colored blocks are the isolates sequenced in this study and were either identified as *C. manginecans* or *C. eucalypticola* as part of the *C. manginecans* complex. Bootstrap values are presented above branches.

**Figure 3 jof-07-00759-f003:**
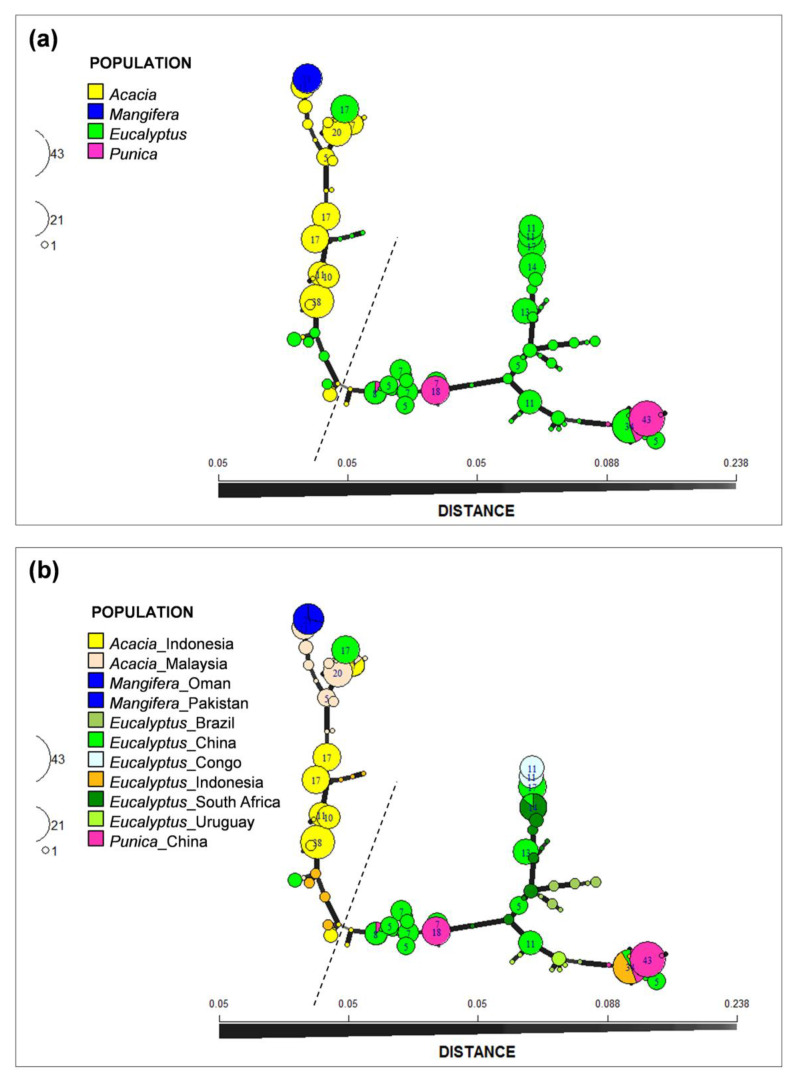
Minimum spanning network constructed using Bruvo’s distances showing two distinct clusters separated by a large genetic distance. The sizes of the nodes are proportional to the number of isolates representing the MLG and the thickness of the lines represent the Bruvo genetic distance between two nodes (thicker lines denote smaller genetic distance). (**a**) Isolates are labeled based on host. (**b**) Isolates are labeled based on host and location. The dashed lines separated the two clusters.

**Figure 4 jof-07-00759-f004:**
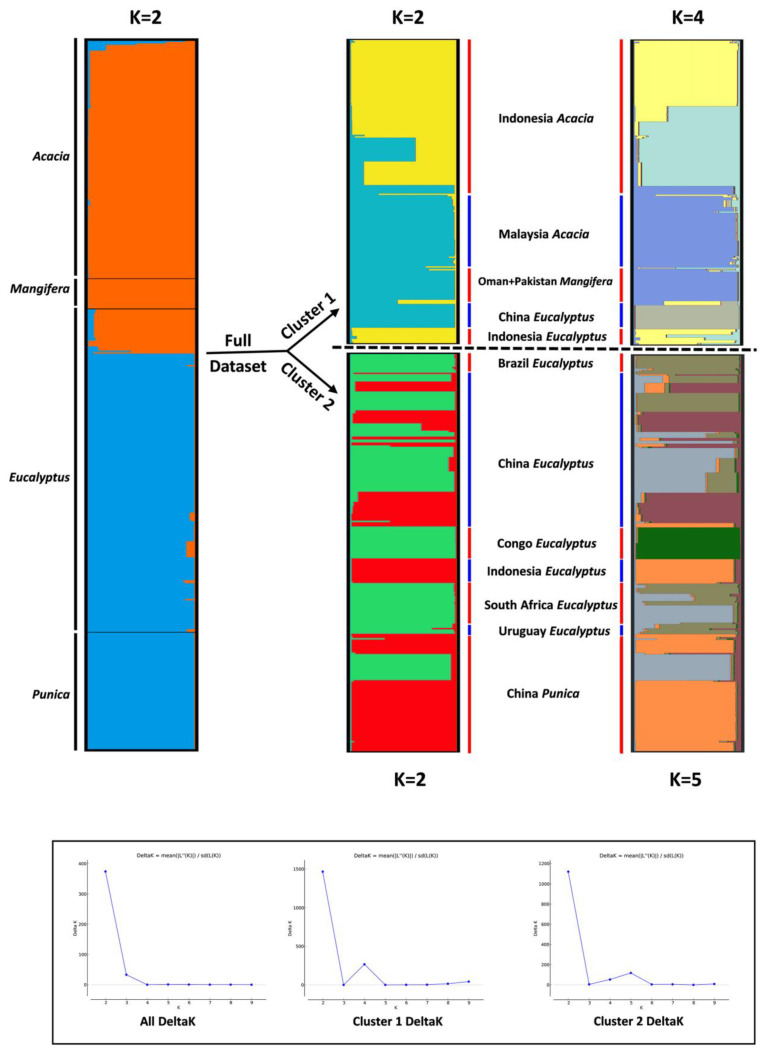
Cluster analyses of isolates in the *C. manginecans* complex inferred using two rounds of hierarchical STRUCTURE analyses. Each individual is represented by a bar, divided into K colors, where K is the possible number of clusters. Arrows delineate the progress of the hierarchical approach, where subsets of the data were subsequently analyzed. For each cluster, ∆*K* is plotted in the bottom.

**Table 1 jof-07-00759-t001:** Genetic diversity statistics for isolates of the *Ceratocystis manginecans* complex grouped into populations based on geographical location and host.

	Population	N ^a^	Na ^b^	Nef ^c^	Hexp ^d^	E ^e^	MLG ^f^	Emlg ^g^	G ^h^
Population based on country	Indonesia (*Eucalyptus* + *Acacia*)	138	26	1.578	0.335	0.705	23	6.290	7.399
	Malaysia (*Acacia*)	55	23	1.240	0.147	0.513	17	5.835	5.243
	Oman (*Mangifera*)	15	10	1	0	0	1	1	1
	Pakistan (*Mangifera*)	6	10	1	0	0	1	1	1
	Congo (*Eucalyptus*)	22	12	1.2	0.1	1	2	2	2
	China (*Eucalyptus* + *Punica*)	207	30	1.671	0.351	0.680	28	7.301	11.528
	Uruguay (*Eucalyptus*)	7	14	1.271	0.160	0.840	5	5	3.769
	Brazil (*Eucalyptus*)	13	20	1.639	0.280	0.770	9	7.692	8.048
	South Africa (*Eucalyptus*)	28	15	1.301	0.176	0.830	10	5.716	4.404
Sub-population based on country and plant host ^i^	Indonesia (*Acacia*)	110	23	1.421	0.250	0.684	14	8.313	5.265
	Indonesia (*Eucalyptus*)	28	21	1.555	0.318	0.496	9	9	2.841
	China (*Eucalyptus*)	126	27	1.760	0.398	0.795	22	13.977	13.121
	China (*Punica*)	81	17	1.273	0.165	0.691	8	4.727	2.730

^a^ N = Number of individuals. ^b^ Na = Number of total alleles observed. ^c^ Nef = Number of effective alleles [46]. ^d^ Hexp = Nei’s unbiased gene diversity [50]. ^e^ E = Evenness. ^f^ MLG = Number of multilocus genotypes (MLG) observed. ^g^ eMLG = The number of expected MLG at the smallest sample size based on rarefaction. ^h^ G = Stoddart and Taylor’s Index of MLG diversity [49]. ^i^ Sub-populations were only divided for China and Indonesia where isolates came from more than one host.

**Table 2 jof-07-00759-t002:** Analysis of molecular variance (AMOVA) of isolates from the *Ceratocystis*
*manginecans* complex from nine geographical populations and four host plant populations.

Populations	Source of Variation	Degrees of Freedom	Sum of Squares	Mean Squares	Estimate of Variance	Percentage of Total Variation (%)	PhiPT Value	*p*-Value ^a^
Geographical origin (Countries)	Among Populations	8	650.64	81.83	1.80	56	0.558	0.001
	Within Populations	482	686.72	1.43	1.43	44	–	0.001
	Total	490	1337.36	– ^b^	3.22	100	–	–
Host plant	Among Populations	3	548.79	182.93	1.71	51	0.513	0.001
	Within Populations	487	788.57	1.62	1.62	49	–	0.001
	Total	490	1337.36	–	3.32	100	–	–

^a^ Levels of significance were based on 999 random permutations. ^b^ “–” represents data that are not available.

**Table 3 jof-07-00759-t003:** Population differentiation of all the *Ceratocystis* populations separated based on country and host. Pairwise comparison of population differentiation (φPT) and gene flow (Nm) among the populations are presented below and above the diagonal, respectively.

	Congo (*Eucalyptus*)	South Africa (*Eucalyptus*)	Uruguay (*Eucalyptus*)	Brazil (*Eucalyptus*)	China (*Eucalyptus*)	China (*Punica*)	Indonesia (*Eucalyptus*)	Indonesia (*Acacia*)	Malaysia (*Acacia*)	Oman (*Mangifera*)	Pakistan (*Mangifera*)
Congo (*Eucalyptus*)	-	0.286	0.237	0.284	0.884	0.213	0.335	0.205	0.115	0.045	0.061
South Africa (*Eucalyptus*)	0.637	-	0.449	0.754	3.209	0.421	0.452	0.232	0.113	0.072	0.093
Uruguay (*Eucalyptus*)	0.679	0.527	-	0.788	2.305	0.526	0.720	0.228	0.110	0.030	0.056
Brazil (*Eucalyptus*)	0.637	0.399	0.388	-	1.416	0.454	0.614	0.265	0.132	0.084	0.131
China (*Eucalyptus*)	0.361	0.135	0.178	0.261	-	1.671	1.631	0.455	0.294	0.272	0.300
China (*Punica*)	0.701	0.543	0.487	0.524	0.230	-	1.360	0.238	0.101	0.079	0.088
Indonesia (*Eucalyptus*)	0.599	0.525	0.410	0.449	0.235	0.269	-	0.601	0.190	0.159	0.210
Indonesia (*Acacia*)	0.709	0.683	0.687	0.654	0.524	0.677	0.454	-	0.478	0.328	0.368
Malaysia (*Acacia*)	0.813	0.815	0.820	0.791	0.630	0.831	0.725	0.511	-	0.737	0.937
Oman (*Mangifera*)	0.917	0.875	0.943	0.856	0.648	0.864	0.758	0.604	0.404	-	-
Pakistan (*Mangifera*)	0.891	0.843	0.899	0.792	0.625	0.850	0.705	0.576	0.348	-	-

## Data Availability

Data are contained within the article and Appendix A.

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
