# Peer review of "Population Diversity and Genetic Structure Reveal Patterns of Host Association and Anthropogenic Impact for the Globally Important Fungal Tree Pathogen *Ceratocystis manginecans"

_jof, 2021, doi:10.3390/jof7090759_

Round 1
Reviewer 1 Report
The paper analyses the genetic structure of a fungal species complex of importance due to its pathogeneicity. The study is well designed though the selection of isolates is not too geographically balanced, so general conclusions are difficult to be drawn. However, some interesting results are found. In my opinion, the most interesting one is the fact that it seems that Eucalyptus might have been a vector of dispersing the pathogen around the world, not only affecting local plantations of this genus but also other genera. This is an important finding in terms of forest health and could be more explicitly expressed in the title of the paper. I have not found any major flaw neither in the analyses, nor in the elaboration of the manuscript. Nevertheless I have the following minor comments:
1st paragraph of the section 2.1: Please add the figure of the total number of isolates used in the study.
3rd parag. section 2.2: Did you used outgroups? It seems so. Which ones?
2nd parag. section 2.3: In section 3.2 it is stated that for some alleles just one isolate was sequenced. Please, be consistent.
1st parag. section 2.4: You did an apriori grouping. That’s not wrong but it might bias the analyses. However, that grouping was consistent with the clustering in sections 2.6 and 2.7. To make things more fluid, and logically sound, I would move sections 2.6 and 2.7 above, before section 2.4 and so make it clear that your clustering is consistent with the structure of the population.
Section 2.7: The runs on Structure should be done “clone-corrected”, ie. Not including in the analyses replicates of the same clone (so not including the 491 isolates).
2nd parag. section 3.1: This paragraph reads like a M&M paragraph. Consider moving it to M&Ms.
1st parag. section 3.2: It would be very illustrative adding estimations of observed heterozygosity and HW disequilibrium.
Figure 2. The clade of C manginecans clomplex seems to be not monophyletic. Any explanations about that? A brief comment in the text would help to understand the figure.
Author Response
Please see our notes to Reviewer in the attached document "Sept. 10, 2021_Response to Reviewer One.docx".

Reviewer 2 Report
p.6, section 2.8 Mode of reproduction. There were actually 999 randomizations rather than 1000.
p.8, last line. It should read "one MLG was shared" rather than "... were shared".
p.10, line 6 from top. Delete "of this".
p.10, line 9. Delete comma after "(Table 3)".
p.10, 2nd paragraph, line 6. Should that read "followed by Eucalyptus"?
p.10, last line. "of the C. manginecans complex"
p.11, Table 2. Should read "Degrees of freedom", not "Degree ..."
p.15, Figure 4 caption. Should that be "Arrows" instead of "Arrow"?
p.15, last line. Should that read "is plotted at the bottom"?
p.16, line 4. A comma is not needed after "hosts".
p.16, line 6. "lastly" not "lastely".
p.16, 3rd paragraph. In one place, "Eucalyptus" is incorrectly spelled "Eucalypstus".
p.17, first complete paragraph. "Moreover, that some introductions have apparently occurred recently." is a sentence fragment.
p.17, Supplementary Materials. Should the two mentions of Ceratocystis be in italics?
p.18, Data Availability Statement. It should read "Data are ..."
Author Response
Please see our notes to Reviewer in the attached document "Sept. 10, 2021_Response to Reviewer Two.docx".
